# Adoption and Implementation Barriers for Worksite Health Programs in the United States

**DOI:** 10.3390/ijerph182212030

**Published:** 2021-11-16

**Authors:** Marc Weinstein, Kalila Cheddie

**Affiliations:** Department of Global Leadership and Management, Florida International University, Miami, FL 33199, USA; kcheddie@fiu.edu

**Keywords:** workplace health promotion, workplace intervention, behavioral health

## Abstract

Worksite health promotion programs have been identified as having the potential to mitigate chronic health risks. In the most recent 2017 U.S. CDC survey of workplace health promotion, respondents identified several perceived barriers related to program adoption and implementation. The analysis indicates that challenges negatively associated with having worksite program were lack of senior management support (OR = 0.50, 95% CI: 0.32–0.78), lack of qualified vendors (OR = 0.56, 95% CI: 0.4–0.79), lack of qualified personnel (OR = 0.56, 95% CI: 0.35–0.73), and cost (OR = 0.58, 95% CI: 0.39–0.88). Challenges associated with having a program were lack of employee interest (OR = 2.09, 95% CI = 1.44–3.03), lack of space (OR = 1.76, 95% CI: 1.26–2.48), and demonstrating program results (OR = 2.09, 95% CI = 1.44–3.03). These findings can provide insights to policy makers, insurers, and employers seeking to implement workplace-based health promotion initiatives.

## 1. Introduction

The tragic impact of the 2020–21 COVID-19 pandemic notwithstanding, social and scientific advances have contributed to steady increases in human longevity. These improvements are observed, despite broad environmental and behavioral changes that have contributed to the obesity epidemic and the rise of metabolic syndrome among Americans [1]. The U.S. Center for Disease Control (CDC) reports that at least six in ten Americans now have a chronic condition [2].

American worksites offer an important venue for health promotion. Over sixty percent of the U.S. population is in the workforce, and Americans spend more time at work than their peers in most industrialized countries [3]. Elements of the Affordable Care Act encouraged employers to offer worksite programs, and workplace health promotion (WHP) programs are now an integral part of the U.S. public health strategy [4,5]. Related to this, self-insurance plans offer employers the opportunity to retain savings, resulting from lower employee health costs, conceivably providing additional incentive for American employers to introduce WHP programs [6].

Since the first administration of this CDC-sponsored survey on WHP in 1985, there has been a steady, but undramatic, increase in the number of scope and WHP programs offered. The CDC’s Healthy People 2000 report defined a comprehensive workplace health promotion program as having five elements: (1) health education programs, (2) supportive social and physical work environment, (3) integration into the organization, (4) linkages to related programs, and (5) health screening and follow up [7]. By this measure, the number of firms offering comprehensive programs increased from 6.9 to 17.1 percent, but this headline statistic may overstate the impact of WHP programs, since many worksites only report having passive, information-only programs and employee participation remains low in many enterprises.

The aim of this paper is to explore the perceived barriers to WHP programs, based on data from the most recent administration of the CDC WHP survey. The cross-sectional data in the workplace health administration survey (WHA) limits our ability to test specific hypothesis. However, the most recent iteration of this survey is of sufficient scope and importance, and it can provide guidance for future research, as well as insights for practitioners.

Previous analyses have offered a snapshot of the prevalence of WHP initiatives nationally [7] or for specific industries, such as healthcare [8]. Other analyses have explored specific areas of behavioral change, such as sleep [6]. To the extent that barriers have been identified, they are typically related to those factors that lead an enterprise to begin to adopt a WHP. Further insights can be gained by examining patterns among perceived barriers for firms that have introduced a WHP program (adopters) and those that have not (non-adopters).

## 2. Materials and Methods

### 2.1. Study Design and Sample

The 2017 WHA survey analyzed here is the most recent iteration of a nationally representative, cross-sectional survey of WHP practices, with previous surveys conducted in 1985, 1992, 1999, and 2004. The sample drew from 2.5 million private and public worksites in the United States with at least ten employees. The data set and data dictionaries are publicly available for download at https://www.cdc.gov/workplacehealthpromotion/data-surveillance/index.html (accessed on 28 August 2021). The stratification criteria for the sample were CDC region, industry, and worksite size. A trained interviewer called each worksite and recruited the person “most knowledgeable about employee health and safety at the worksite”. The respondent typically spent 40 min with the interviewer to provide information about practices at that worksite. The survey was administered between November 2016 and September 2017, and the response rate was 10.1% (*n* = 3109).

### 2.2. Measures

Worksite characteristics in the analysis include firm size, industry, and percent of worksite represented by a union. The raw data from WHA provides categories for firms as small as 10–24 employees and includes categories for firms between 500–750 employees, as well as for those greater than 750. There are many ways to compare worksites by size. For the purposes of this analysis, we follow the lead of the original data gathering team [7] and collapse the two largest size categories into one category (500+ employees). We retain the original firm industry designations used by Linnan et al. [7]. To represent unionization, the union variable was coded 1 for companies where 10 or more percent of its employees are collectively represented. Table 1 contains frequencies for these variables. The WHA survey categorized establishments into seven industry groups. Table 2 contains the frequencies for these variables.

The WHA includes nine questions, asking respondents whether a range of factors pose a barrier to having a WHP (1 = not all challenging, 2 = slightly challenging, 3 = somewhat challenging, 4 = challenging, and 5 = extremely challenging). For each, we created a dummy variable by coding “challenging”, and “extremely challenging” as 1 and the other values as zero. Table 3 provides the frequencies of companies identifying specific barriers as “challenging” and “extremely challenging”.

To measure the adoption of WHP, the WHA first asks respondents whether they have had any WHP initiatives in the last 12 months. Subsequently, establishments that responded “yes” were asked about nine WHP different initiatives. These are listed in Table 4, together with frequencies, based on the entire sample.

### 2.3. Analysis

To explore the relationship between the identified barriers and the adoption of a WHP, we conducted a logistic regression routine in SPSS version 27, in which the dependent variables were the adoption of a WHP initiative. The first step in the data analysis strategy was to model the adoption of any WHP initiative as a function of firm size, firm industry, unionization status, and barriers to adoption. This is accomplished in a stepwise fashion, in order to explore the added explanatory power of each group of firm characteristic variables. The WHA survey has additional data about firm characteristics, including workforce demographics. These variables were found to not be statistically associated with WHP, did not improve model fit, and are not included in the final analysis. For worksite size, the smallest enterprises (10–24 employees) are in the base of the equation, i.e., this is the level of the variable that is compared to the other levels of the variable. In the case of the industrial sector, Industry 1 (agriculture, mining, construction, and manufacturing) is in the base of the equation.

Once we established the relative explanatory power of each group of variables, we used a fully specified model to explore patterns for each of nine WHP initiatives: (1) physical activity, (2) nutrition, (3) obesity reduction, (4) tobacco cessation, (5) alcohol abuse, (6) lactation support, (7) musculoskeletal disease (MSD) reduction, (8) stress management, and (9) sleep management.

## 3. Results

In Models 1–4 (Table 5), the dependent variable is dichotomous and reflects whether the responding company had any WHP initiative in the last 12 months. Model 1 includes dummy variables to indicate firm size, with the category 10–24 employees in the base of the model. Model 2 adds industry variables, with the establishments in the agricultural, mining, and construction (one category in the WHA survey) being in the base of the model. Model 3 adds Union, and the final model has all previously included variables and 12 measures of perceived barriers. The log-likelihood ratios associated 95% confidence intervals, and Nagelkerke pseudo R-square estimates for Models 1–4 are provided.

The number of employees at a worksite is positively associated with the likelihood of a worksite having a WHP program. Industry differences were also observed, with hospitals (OR = 3.24; 95% CI, 1.77–5.95, *p* < 0.001) and public sector worksites (OR = 3.08; 95% CI, 1.69–5.60, *p* < 0.001) more likely to have worksite health programs than worksites in other industries. In worksites, in which at least 10 percent of the employees are covered by the collective bargaining agreement, the existence of a collective bargaining agreement for at least 10 percent of employees is positively associated with the adoption of worksite health programs within the past 12 months (OR = 3.38; 95% CI, 2.18–5.29, *p* < 0.001).

With regards to perceived barriers, the variables broadly group into two areas. First, there are those variables that are negatively associated with a worksite having a health program. These include lack of top leadership support (OR =.50; 95% CI, 0.32–0.78, *p* < 0.01), cost (OR = 0.58; 95% CI, 0.39–0.88, *p* < 0.01), lack of personnel (OR = 0.51; 95% CI, 0.35–0.74, *p* < 0.001), and lack of vendors (OR = 0.56; 95% CI, 0.40–0.79, *p* < 0.001). Others are barriers that are positively associated with worksite programs. These are: lack of employee interest (OR = 2.09; 95% CI, 1.44–3.03, *p* < 0.001), the need to demonstrate results (OR = 2.09; 95% CI, 1.44–3.03, *p* < 0.001), and lack of space (OR = 1.76; 95% CI, 2.47, *p* < 0.001).

Table 6 uses the full model specification (Model 4 in Table 5) when estimating comparable parameters for each of the WHPs. Establishment size is positively associated with the likelihood of offering individual WHP initiatives. There are some interesting nuances when examining specific programs. For obesity reduction, for instance, the distinction is between firms with 500 or more employees and all other establishments. In other areas that are more universal, for example, physical activity and nutrition, the likelihood of an establishment having a WHP increase with each size category. This is in contrast to newer areas of interest, such as stress management and sleep management, in which the association between size and likelihood of having a WHP initiative is weaker.

## 4. Discussion

The positive association between the number of employees at the worksite and WHP initiatives likely reflects the economies of scales intrinsic to these programs and the better benefits packages typically offered by larger enterprises [9]. Similarly, hospitals are more likely to have worksite health programs, due to both the support culture of these worksites, as well as internal expertise that effectively reduce the marginal expense of these programs [8]. The higher likelihood of having WHP in the public sector may also have an economic explanation. The longer tenure of public sector employees increases the incentive to address chronic diseases associated with diet and lifestyle, since they are delayed financial returns to effectively addressing chronic disease risk [9]. The observed association between unionization and WHP is consistent with the better benefits typically enjoyed by employees covered by collective bargaining agreements [10].

Interesting patterns also emerge when examining the industry. Two sectors, public administration and healthcare services, stand out as having a higher likelihood of WHP initiatives, except for three types of programs—alcohol, sleep management, and MSD risk. In the case of alcohol, the most probable explanation is that support for employees suffering from alcoholism is among the old health interventions in the United States [11]; as such, these programs are widely dispersed. In the area of sleep management, there is no industry association, reflecting the likely novelty of such programs. In the area of MSD, there is a small, but statistically significantly higher likelihood of establishments in the arts and technical services. Less discussed in the literature is the relationship between public administration establishments and WHP initiatives. This is an area that requires further investigation, with likely explanations including the better benefits packages these employees enjoy (in part to offset lower compensation); the longer tenure of these employees suggests there could be downstream savings from enterprise behavioral interventions.

The association between union coverage and the likelihood of an establishment having a specific WHP initiative is strong for each of the nine programs covered in the WHA survey. Firms with at least 10 percent of employees covered by a union contract are estimated to be 2.5 to 4.8 times more likely to have programs with less or no union coverage. This reflects a number of long-term dynamics in the United States that may contribute to this. First, firms with unions in the United States tend to have better benefits, as employers have better benefits packages than their counterparts in establishments without unions [12]. While unions may only cover groups of non-managerial employees, the benefits that unions bargain for their members are then often extended to non-unionized employees throughout the establishment. As might be the case with public sector employees, there could be downstream savings from enterprise behavioral interventions.

When examining the relationship between perceived barriers to the likelihood of an establishment having a specific WHP initiative, a clear pattern emerges between perceptions of barriers related to whether an establishment decides to launch a program, as well as the perceived barriers that exist once a company has a program. For instance, variables associated with program implementation, lack of employee interest, lack of middle management support, and the need to demonstrate results are all associated with establishments having WHP initiatives. By contrast, when an establishment lacks top management support, it is concerned with cost and notes the lack of personnel and vendors: these establishments are less likely to have WHP initiatives than other establishments controlling for firm characteristics. Notably, perceived barriers related to confidentiality and employee distrust were not statistically associated with the likelihood of an establishment having a WHP initiative.

## 5. Conclusions

The 2017 WHA survey indicates that, despite policies designed to promote them, WHP programs still have limited diffusion. Despite the inherent limitations of the cross-sectional data and limited information about worksite characteristics, important patterns emerge. The perceived leadership support, costs, limited personnel, and expertise in this area are identified as barriers in worksites that have already launched these programs. Once initiated, practitioners are more likely to be concerned with demonstrating the results of these programs and with overcoming lack of interest of employees. As challenging as these barriers may be, they are tractable issues that can be overcome in the promotion and success of WHPs.

The 2020–21 COVID pandemic has, for many U.S. establishments, shifted the focus from WHPs initiatives to remote work and vaccination policies. Nonetheless, there is every indication that the challenge of behavioral health and chronic disease will remain. In the United States, where employer-based health insurance is the norm, the workplace will continue to offer opportunities for health interventions.

For companies seeking to introduce new WHPs, this analysis suggests that concerns about costs, vendors, and the absence of management support are impediments to WHP implementation. For these establishments, evidence of cost savings from WHP initiatives, the growth of WHP options, and supportive public policy may encourage more companies to launch WHP programs. This analysis also indicates that gaining middle management support and demonstrating results are important for sustaining WHP efforts. The findings from this analysis also affirm the relationship between establishment size and the likelihood of an establishment having a WHP program. This reflects the economies of scale, often found in many human resource activities, and highlights the need to reduce costs and increase accessibility for medium and small employers.

The WHA survey is intended to provide researchers and practitioners a snapshot of health promotion activities in the United States. The administration of this survey, however, has been irregular, often with long gaps between administration and low response rates (10.1 percent in 2017). Another limiting factor is that the WHA relies on a single respondent, and there are no links to independent measures of performance. Gaining a deeper and more nuanced understanding of the factors contributing the diffusion and success of WHP initiatives requires a multi-year collaborative research endeavor to support a panel study of firms, data about their health promotion activities, employee health outcomes, and health care expenditures.

## Figures and Tables

**Table 1 ijerph-18-12030-t001:** Firm size (percentages), WHA survey, 2017.

Firm Size	Percentage
10–24 Employees	1175
	(41.3)
25–49 Employees	654
	(23.0)
50–99 Employees	364
	(12.8)
100–249 Employees	265
	(9.3)
250–499 Employees	131
	(4.6)
500+ Employees	254
	(8.9)

**Table 2 ijerph-18-12030-t002:** Industry and unionization (percentages), WHA survey, 2017.

Industry	Number of Worksites in Sample
Agriculture mining, utility, construction, manufacturing	526
	(18.5)
Trade, retail, transportation, warehouse	310
	(10.9)
Arts, food services	431
	(15.2)
Information, finance, real estate technical, consult, admin, waste	430
	(15.1)
Education, healthcare (excluding hospital works)	552
	(19.4)
Public administration	256
	(9.0)
Hospital worksites—general, surgical, psychiatric, other	338
	(11.9)
Unionized	444
	(15.6)
*n* = 2843	

**Table 3 ijerph-18-12030-t003:** Specific barriers to worksite health promotion, WHA survey, 2017.

Barriers	Percentage
Financial costs	56.4
Competing demands	40.1
Lack of employee interest	39
Lack of trained personnel	32.2
Demonstrating results	25.5
Senior management	20.6
Middle management	17.9
Concerns about confidentiality	15.6
Employee distrust	13.7
*n* = 1792–2050	

**Table 4 ijerph-18-12030-t004:** Percentage of firms that have programs in the last twelve months, WHA survey, 2017.

Worksite Programs	Percentage
Any program	52.2
Physical activity	31.6
Nutrition	26.3
Obesity	16.3
Tobacco	22.3
Alcohol	15.4
Lactation support	10.7
MSD program	13.1
Stress management	22.8
Sleep management	9.5
*n* = 2843	

**Table 5 ijerph-18-12030-t005:** Logistic regression for worksite programs, odds ratios (CI 95%) (WHA survey, 2017).

	Model 1	Model 2	Model 3	Model 4
25–49 Employees ^1^	1.19 (0.97–1.45)	1.16 (0.948–1.422)	1.14 (0.90–1.44)	1.12 (0.81–1.55)
50–99 Employees	2.24 (1.76–2.87) ***	2.06 (1.60–2.65) ***	1.85 (1.38–2.48) ***	2.29 (1.45–3.49) ***
100–249 Employees	4.14 (3.05–5.62) ***	3.38 (2.46–4.65) ***	3.05 (2.11–4.40) ***	2.68 (1.65–4.37) ***
250–499 Employees	7.27 (4.49–11.78) ***	5.67 (3.16–8.76) ***	5.15 (2.75–9.65) ***	5.63 (2.16–14.68) ***
500+ Employees	15.12 (9.61–23.79) ***	10.36 (6.38–16.81) ***	9.13 (5.07–16.41) ***	7.09 (3.30–15.29) ***
Trade ^2^		1.40 (1.04–1.89) *	1.38 (0.99–1.94)	1.48 (0.92–2.37)
Arts		0.94 (0.71–1.24)	0.91 (0.66–1.24)	1.31 (0.84–2.06)
Tech services		1.18 (0.90–1.54)	1.13 (0.83–1.54)	1.49 (0.97–2.23)
Ed Services		1.66 (1.28–2.14) ***	1.26 (0.94–1.71)	1.44 (0.95–2.18)
Public administration		4.10 (2.86–5.89) ***	2.17 (1.41–3.25) ***	3.08 (1.69–5.60) ***
Hospitals		2.33 (1.59–3.40) ***	2.32 (1.45–3.64) ***	3.24 (1.77–5.95) ***
Union			3.07 (2.27–4.17) ***	3.38 (2.18–5.29) ***
Lack of employee interest				2.09 (1.44–3.03) ***
Employee distrust				1.17 (0.86–1.60)
Business demands				1.16 (0.82–1.65)
Confidentiality				1.09 (0.78–1.15)
Need for results				2.09 (1.44–3.03) ***
Lack of mid man support				1.40 (0.88–2.21)
Legal concerns				1.00 (0.72–1.41)
Lack of space				1.76 (1.26–2.48) **
Lack of top man support				0.50 (0.32–0.78) **
Financial cost				0.58 (0.39–0.88) *
Lack of Personnel				0.56 (0.35–0.73) ***
Lack of qualified vendors				0.56 (0.40–0.79) **
Nagelkerke R-square	0.167	0.206	0.215	0.324

* *p* < 0.05, ** *p* < 0.01, *** *p* < 0.001. ^1^ In the WHA survey, establishments were categorized into six size categories. Establishments with 1–24 employees is the reference group to which the other size categories should be compared. ^2^ In the WHA survey, establishments were categorized in one of seven broad industrial groups. Establishments in the group agricuture, mining, construction, and manufacturing is the reference group to which other industrial categories should be compared.

**Table 6 ijerph-18-12030-t006:** Logistic regression for adoption of worksite health promotion programs (WHA survey, 2017).

	Physical Activity	Nutrition	Obesity	Tobacco	Alcohol	Lactation	MSD	Stress	Sleep
25–49 Employees	0.89 (0.62–1.27)	0.83 (0.56–1.28)	0.68 (0.44–1.05)	0.68 (0.44–1.03)	0.72 (0.46–1.11)	0.97 (0.53–1.76)	0.55 (0.34–0.90) *	0.78 (0.52–1.17)	0.53 (0.30–0.93) *
50–90 Employees	1.25 (0.81–1.92)	1.35 (0.85–2.13)	0.85 (0.44–1.43)	1.10 (0.65–1.77)	1.02 (0.61–1.70)	1.66 (0.89–3.12)	0.82 (0.38–1.41)	1.17 (1.03–2.70) *	0.91 (0.4–1.69)
100–249 Employees	2.63 (1.65–4.19) ***	2.63 (1.65–4.19) ***	1.46 (0.87–2.47)	2.04 (1.25–3.34) **	1.69 (1.00–2.85) *	2.18 (1.18–4.02) ***	1.11 (0.64–1.93)	1.70 (1.03–2.70) *	0.59 (0.30–1.18)
250–499 Employees	3.06 (1.45–6.47) **	3.06 (1.45–6.47) ***	2.57 (1.23–2.66)	1.93 (0.93–3.99)	0.85 (0.38–1.91)	2.47 (1.08–6.66) *	1.66 (0.70–3.56)	1.50 (0.73–3.07)	0.30 (0.10–0.89) *
500+ Employees	3.83 (2.16–6.8) ***	6.81(3.66–12.68) ***	6.81(3.66–12.68) ***	6.16(3.41–11.13) ***	2.56 (1.44–4.56) ***	7.88(4.17–14.89) ***	2.38 (1.32–4.29) **	5.64 (3.10–10.25) ***	1.40 (0.70–2.77)
Trade	1.54 (0.93–2.56)	1.45 (0.84–2.53) *	1.68 (0.95–2.97)	1.37 (0.08–2.36)	1.32 (0.75–2.32)	1.00 (0.67–2.11)	1.19 (0.67–2.11)	1.36 (0.78–2.38)	1.41 (0.70–2.82)
Arts/Entertainment	1.70 (1.05–2.78) *	1.70 (1.05–2.78) *	1.88 (1.11–3.17) *	1.16 (0.65–1.99)	1.28 (0.74–2.22)	1.13 (0.50–2.56)	0.43 (0.22–0.84)	1.54 (0.91–2.63)	0.60 (0.27–1.33)
Tech Services	1.48 (0.93–2.35)	1.62 (0.97–2.68)	1.10 (0.62–1.94)	0.85 (0.5–1.56)	0.71 (0.40–1.25)	1.78 (0.91–3.51)	0.52 (0.28–0.95)	1.39 (0.83.–2.32)	1.28 (0.66–2.46)
Educational Services	1.64 (1.096–2.53) *	2.12 (1.33–3.37) ***	0.92 (0.54–1.58)	0.96 (0.50–1.56)	0.74 (0.44–1.25)	1.88 (1.01–3.51) *	0.75 (0.45–1.27)	1.70 (1.06–2.73) *	0.86 (0.45–1.65)
Public Admin	3.72 (2.11–6.54) ***	3.52 (1.97–6.29) ***	2.66 (1.49–4.76) ***	2.45 (1.30–4.33) ***	1.58 (0.89–2.80) ***	1.47 (0.68–3.16)	1.23 (0.60–2.21)	2.96 (1.60–5.23) ***	1.63 (0.85–3.19)
Health Services	2.68 (1.54–4.65) ***	2.57 (1.44–4.58) ***	2.73 (1.54–4.85) ***	1.78 (1.00–3.15) *	1.08 (0.59–1.97)	4.03 (2.07–7.86) ***	1.24 (0.68–2.25)	3.14 (1.78–5.54) ***	1.76 (0.85–3.66)
Union	3.43 (2.33–5.04) ***	3.56 (2.41–5.25) ***	2.80 (1.89–4.14) ***	2.55 (1.73–3.77) ***	4.04 (2.76–5.93) ***	1.94 (1.20–3.12) ***	4.22 (2.85–6.23) ***	3.74 (2.55–5.49) ***	4.86 (3.14–7.52) ***
Lack of interest	2.67 (1.77–4.08) ***	2.71 (1.72–4.58) ***	2.54 (1.54–4.19) ***	3.51 (2.13–5.78) ***	2.62 (1.59–4.34) ***	1.28 (0.60–2.36)	1.58 (0.965–2.62)	2.25 (1.43–3.54) ***	2.65 (1.38–5.08) **
Employee distrust	1.01 (0.74–1.39)	1.38 (0.98–1.93)	1.07 (0.74–1.54)	1.06 (0.75–1.50)	1.16 (0.81–1.68)	1.29 (0.82–2.03)	1.34 (0.91–1.97)	1.11 (0.79–1.57)	1.49 (96–2.32)
Competing demands	1.37 (0.96–1.97)	1.05 (0.72–1.54)	0.91 (0.61–1.35)	1.10 (0.74–1.64)	0.92 (0.62–1.38)	1.89 (1.11–3.20) *	1.14 (0.74–1.75)	0.96 (0.66–1.41)	1.54 (0.93–2.58)
Confidentiality	1.02 (0.73–1.42)	1.14 (0.80–1.62)	0.74 (0.51–1.07)	0.93 (0.65–1.64)	0.75 (0.51–1.10)	1.75 (1.10–2.78) *	.80 (0.51–1.19)	0.96 (0.57–1.36)	1.01 (0.64–1.59)
Need for results	2.45 (1.68–3.61) ***	2.79 (1.85–4.21) ***	2.32 (1.50–3.59) ***	2.62 (1.71–4.01) ***	1.52 (0.99–2.34)	1.29 (0.74–2.22)	1.67 (1.06–2.63) *	2.39 (1.58–3.61) ***	1.59 (0.93–2.71)
Lack of mid man sup	1.15 (0.73–1.82)	1.69 (1.04–2.75) *	1.63 (0.98–2.70)	1.98 (1.21–3.27) ***	1.74 (1.05–2.89) *	1.05 (0.56–1.89)	1.85 (1.0–3.13)	2.22 (1.36–3.64) **	2.92 (1.61–5.32) ***
Legal concerns	1.18 (0.84–1.66)	1.05 (0.73–1.51)	1.14 (0.78–1.67)	1.42 (0.98–2.06)	1.28 (0.88–1.87)	1.21 (0.76–1.93)	1.26 (0.84.1.88)	1.21 (0.88–1.74)	0.95 (0.60–1.52)
Lack of Space	1.52 (1.07–2.15) *	1.46 (1.01–2.11) *	1.69 (1.13–2.52)	1.22 (0.83–1.79)	1.52 (1.0–2.26) *)	1.16 (0.71–1.88)	1.05 (0.70–1.58)	1.27 (0.88–1.85)	0.87 (0.54–1.38)
Lack of top man sup	0.54 (.34–0.84) **	0.41 (0.30–0.80) ***	0.49 (0.30–0.80) ***	0.46 (0.20–0.75) **	0.52 (0.20–0.85) **	0.72 (0.39–1.22)	0.28 (0.17–47) ***	0.46 (28–0.74) ***	0.34 (0.19–0.61) ***
Financial Cost	0.52 (0.34–79) ***	0.59 (0.38–0.49) ***	0.91 (0.56–1.47)	0.57 (0.36–0.90) *	1.01 (0.62–1.63)	0.49 (0.27–0.87) *	1.07 (0.65–1.75)	0.54 (0.35–0.85) **	0.45 (0.26–0.78) **
Lack of personnel	0.46 (0.32–0.66) ***	0.33 (0.23–0.76) ***	0.40 (0.27–0.58) ***	0.35 (0.24–0.52) ***	0.35 (0.24–0.52) ***	0.75 (0.46–1.24)	0.58 (0.38–0.88) *	0.52 (0.35–76) ***	0.51 (0.32–0.83) **
Lack of vendors	0.53 (0.37–0.74) ***	0.53 (0.37–0.76) ***	0.59 (0.41–0.87) **	0.44 (0.30–0.63) ***	0.64 (0.44–0.94) *	0.60 (0.38–0.95) *	0.77 (0.52–1.15)	0.47 (0.33–0.68) ***	0.55 (0.35–0.87) ***
Nagelkerke R-Sq	0.333	0.389	0.282	0.369	0.188	0.305	0.287	0.350	0.327

* *p* < 0.05, ** *p* < 0.01, *** *p* < 0.001.

## Data Availability

Data download from https://www.cdc.gov/workplacehealthpromotion/data-surveillance/index.html (accessed on 28 August 2021).

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
