# Peer review of "Adoption and Implementation Barriers for Worksite Health Programs in the United States"

_ijerph, 2021, doi:10.3390/ijerph182212030_

Round 1

Reviewer 1 Report

This paper is about the adoption and barriers for worksite health programs. It is interesting but the writing is poor....

  1. there is no background information in the abstract, and no knowledge contribution can be identified from the abstract
  2. page 1, please define non-adopters? why non-adopters are of interest besides adopters?
  3. page 2, line 49, please give the source of the 2017 WHA survey, where to find it?
  4. Table 1, why the firm size is categorized in this way? for example, does 10-24 employees represent small firms? but what about the firm size from 1 to 9, any data about it?
  5. Table 1 is poorly formatted, why not separate it into two tables? one for firm size, the other one for industry
  6. page 3, line 73, using a dummy variable, value 1, to represent challenging and extremely challenging does not make sense to the reviewer. In this case, value 0 covers not all challenging, slightly challenging, and some what challenging. Basically, this misleads the results, because slightly challenging and some what challenging are also challenging, right? but they are missing from "Percentage column".
  7. For table 2, the values of the percentage can be re-ordered from the largest to the smallest.
  8. For table 3, please clarify what is "Any program"? and where are the 12 WHP different initiatives? there are only 9 there.
  9. page 4, line 98, what is "Industry 1"? in the model,  are the values of the four sectors including agriculture, mining, construction and  manufacturing combined or separated?
  10. page 4, line 106, "models 1a-1d" should be changed to models 1 -4, right? Why model 1b or model 2 agricultural sector is the base, where are the construction, mining, and manufacturing sectors?
  11. in Table 4, what is "Results"?
  12. Table 5, it is not possible to read it....please also make sure the row names are accurate. For example, "Cost" should be "Financial cost" in Table 5. 
  13.  

Reviewer 2 Report

Suggestions for the authors to consider:

Introduction

  • When did the most recent CDC survey occur? Even though stated in “Materials and Methods,” it would be helpful to state in the introduction.

  • Do the authors have any hypotheses to be tested?

Discussion

  • Do the authors have suggestions to increase the implementation and utilization of WHPs?

  • What further research should be done in this area?

Author Response

Attached response to Reviewer #2

Reviewer 3 Report

Thank you for the opportunity to review this paper, where the authors explore barriers to worksite health promotion programs across a large sample of US firms. This paper has a promising premise, and does a good job at interrogating the data available to the authors to draw conclusions. However, there is a lack of that next step between findings and recommendations, and the paper suffers from a lack of clarity in several key aspects. I would recommend thorough revision of the work to address some confusing and/or unclear aspects.

  1. I found the third paragraph (beginning “Since the first…”) slightly confusing. The five elements of the Healthy People 2000 report (which, to enrich the paper, should be listed here) does not seem to have any clear link to the theme of “increase in the number of scope and number of WHP programs offered” set out at the start of this paragraph; then mentioned is “an increase from 6.9 percent to 17.1 percent” – but of what? Of firms offering a WHP program that meets this criteria? And is this comparing 1985 to 2000, or to 2020?
  2. In Section 2, it is unclear who the survey was distributed to specifically. To individuals at an employee level (i.e. anyone can answer about the organisation they work for)? Or at an organisation level (i.e. surveys are distributed to organisations/worksites, and were answered by a representative of the organisation/worksite on the organisation’s behalf?)
  3. The Percentages in Table 1 do not add to 100 – the first section is 99.9 (presumably just a rounding issue), but the Industries section adds to 115.6%. Presumably this is overlap as some firms consider themselves as active in more than one sector? However, this should be indicated if it is the case.
  4. I cannot comment on the validity of the quantitative analysis, as this is outside my specialty area. However, I would recommend removal of the qualitative suppositions made to explain findings (e.g. line 138-9) without any basis from the data to make them. Table 5 also has a “placeholder” header rather than an accurate descriptive header.
  5. The Discussion of the findings is mostly clear; although there are several generalised statements made without any citations to support them (e.g. line 179-86 regarding the way unions work within organisations to affect things like health initiative)
  6. It is a significant oversight not to include a “Practice/Policy implications” section. While the findings are themselves interesting, how do they translate into recommendations link up or take the next step to addressing the issues introduced at the start? Where can they feed into, at a policy or practice level, and what kind of organisational/governmental partners might need to be involved to put recommendations arising from the findings into practice?
  7. It is a significant oversight not to include a “Limitations and future research” section. There are several limitations that should be considered, and could be mentioned, here. The major one is that this quantitative work, while strong, cannot address the “why” questions that qualitative work does. For example, why do the HR departments of firms in particular sectors perceive need for or resistance to these sorts of initiatives/programs? Another example, what effect does the lack of programs have on the Quality of Work Life or experience of work for employees? Such qualitative work is beyond the scope of this study, but should be acknowledged as a gap in the knowledge that this study cannot address. Other limitations could include language: i.e. this survey may have excluded small migrant-run firms where the owner/CEO and employees, and the clients/service users too, do not have English as a first language, and who largely operate in a language other than English; how would any practice/policy implications of the findings reach firms like this?

Author Response

Attached response to Reviewer #3.

Round 2

Reviewer 3 Report

I commend the authors for their thorough and thoughtful revisions and response to reviewer comments. All my comments and suggestions have been well addressed in revisions. I have only one new comment to make, regarding the revised Abstract: The new sentences at the start are good but now there is something missing in connecting the problem (first two sentences) with the findings – needs one other sentence stating what this study did (v. brief methods description in 1 sentence e.g. "This study analysed elements of the 2017 Workplace Health Administration Survey to identify these barriers".)

Author Response

We have re-written the abstract as recommend. Than you for this recommendation and your earlier thoughtful comments on our manuscript.